# Novel *ATP2A2* Gene Mutation c.118G>A Causing Keratinocyte and Cardiomyocyte Disconnection in Darier Disease

**DOI:** 10.3390/biomedicines12051060

**Published:** 2024-05-10

**Authors:** Andrea Frustaci, Alessandro De Luca, Romina Verardo, Valentina Guida, Maria Alfarano, Camilla Calvieri, Luigi Sansone, Matteo Antonio Russo, Cristina Chimenti

**Affiliations:** 1IRCCS L. Spallanzani, Cellular and Molecular Cardiology Laboratory, 00149 Rome, Italy; romina.verardo@inmi.it; 2IRCSS Fondazione Casa Sollievo della Sofferenza, Medical Genetics Division, 71013 San Giovanni Rotondo, Italy; a.deluca@css-mendel.it (A.D.L.); v.guida@css-mendel.it (V.G.); 3Department of Cardiovascular, Respiratory, Nephrologic, Anesthesiologic and Geriatric Sciences, Sapienza University of Rome, 00185 Rome, Italy; maria.alfarano@uniroma1.it (M.A.); camilla.calvieri@uniroma1.it (C.C.); cristina.chimenti@uniroma1.it (C.C.); 4IRCCS San Raffaele, 00166 Rome, Italy; luigi.sansone@sanraffaele.it (L.S.); matteoantoniorusso44@gmail.com (M.A.R.); 5Department of Human Sciences and Promotion of Quality of Life, San Raffaele Open University, 00166 Rome, Italy

**Keywords:** Darier disease, gene variant, SERCA2, cardiomyocyte disconnection, molecular and cellular rehabilitation

## Abstract

Darier disease (DD) is an autosomal dominant disorder due to pathogenic variants of the *ATP2A2* gene that causes an isolated skin manifestation based on keratinocyte disconnection and apoptosis. Systemic manifestations of DD have not been demonstrated so far, although a high incidence of neuropsychiatric syndromes suggests an involvement of the central nervous system. We report that the pathogenic ATP2A2 gene variant c.118G>A may cause cardiac involvement in patients with DD, consisting of keratinocyte and cardiomyocyte disconnection. Their common pathologic pathway, still unreported, was documented by both skin and left ventricular endomyocardial biopsies because cardiac dilatation and dysfunction appeared several decades after skin manifestations. Keratinocyte disconnection was paralleled by cardiomyocyte separation at the lateral junction. Cardiomyocyte separation was associated with cell disarray, sarcoplasmic reticulum dilatation, and increased myocyte apoptosis. Clinically, hyperkeratotic skin papules are associated with chest pain, severe muscle exhaustion, and ventricular arrhythmias that improved following administration of aminophylline, a phosphodiesterase inhibitor enhancing SERCA2 protein phosphorylation. Cardiac pathologic changes are similar to those documented in the skin, including cardiomyocyte disconnection that promotes precordial pain and cardiac arrhythmias. Phosphodiesterase inhibitors that enhance SERCA2 protein phosphorylation may substantially attenuate the symptoms.

## 1. Introduction

Darier disease (DD) is an autosomal dominant disorder caused by pathogenic variants of the *ATP2A2* gene, located on chromosome 12q23-24.1 and encoding for sarco/endoplasmic reticulum Ca^++^ ATPase type 2 isoform (SERCA2). The gene transcript produces two isoforms (a and b) with either 21 or 20 exons, respectively. Both isoforms contain phosphorylation, nucleotide-binding, and actuator domains. According to the LOVD database (https://databases.lovd.nl/, accessed on 1 March 2024), there are 276 unique pathogenic variants reported for *ATP2A2*, with no prominent variant hotspots. These variants are predominantly missense/nonsense alterations, along with deletions, splice sites, and insertions. Previous studies of different *ATP2A2* variants (i.e., missense, nonsense, and deletions) have shown that they all affect SERCA2 function by either decreasing protein expression, Ca^++^-ATPase activity, and Ca^++^ transport, or altering protein kinetic properties. This disruption leads to impaired or lost Ca^++^ transport in the ER, thereby reducing ER Ca^2+^ stores. The resulting imbalance in Ca^++^ affects skin cell adhesion, potentially contributing to the DD skin phenotype. Clinically, DD manifests with hyperkeratotic papules in seborrheic areas, palmo-plantar pits, and distinctive nail dystrophy. Histologically, it is characterized by loss of adhesion and apoptosis of keratinocytes (acantholysis) [1,2].

Consistent extra-cutaneous manifestations of Darier disease have not been documented so far [2], and specifically, the heart has been reported as non-affected [3].

In the present study, a novel pathogenic variant of *ATP2A2* is reported, severely affecting the skin and the myocardium. Clinical manifestations are supported by histological evidence of keratinocyte and cardiomyocyte disconnection. Clinical symptoms of muscle exhaustion, chest pain, and cardiac arrhythmias have been attenuated by phosphodiesterase inhibitor administration.

## 2. Materials and Methods

A 62-year-old lady was admitted because of palpitations and chest pain at rest, increasing with effort and limiting remarkably the quality of life, resulting in a depressive syndrome with a failed attempt of suicide. She was affected since the age of nine by Darier disease (Figure 1), manifesting with extensive greasy, crusted, yellow–brown papules on the seborrheic areas. Beyond skin lesions (Figure 2), clinical examination was unremarkable. She was afebrile, well nourished, with a body weight of 65 kg, and her blood pressure was normal. Beyond Darier disease manifestations, her anamnesis was unremarkable. In particular, concomitant diseases like diabetes, alcoholism, drug addiction, and other chronic pathologies were not reported. In the last two years, she developed cardiac symptoms such as chest pain, palpitations, and dyspnoea on effort that required cardiological investigations.

Routine blood and chemical tests were within normal limits. Indices of inflammation like erythrocyte sedimentation rate and C reactive protein were normal. ECG showed sinus rhythm at 54 bts/min with abnormal ST segment deflection (−1, −1.5 mm) in the infero-lateral leads and negative T wave (Figure 3).

### 2.1. Cardiac Studies

We performed non-invasive (stress ECG, Holter monitoring, 2D-echo, and magnetic resonance) and invasive investigations (coronary angiography and left ventricular endomyocardial biopsy) after written informed consent. Specifically, five endomyocardial samples were drawn from the left ventricular septum and processed for histology and electron microscopy, as previously described [4]. Apoptosis of cardiomyocytes was assessed by a hairpin probe as described [4]. Morphometric evaluation of unendotheliolized myocardial spaces as a result of cardiomyocyte disconnection has been obtained in 10 sections of 3 biopsy fragments and compared with normal controls (2F, 1M mean age 60.6 ± 1.15 years, under no medication, undergoing surgical coronary artery grafting in normal cardiac parameters and function).

Cardiomyocyte apoptosis was evaluated by in situ ligation of hairpin probes with single-base 3′ overhangs (hairpin 1) or blunt ends (hairpin 2).

### 2.2. Skin Biopsy

It was obtained from an affected skin area after local anesthesia. It was used to confirm the histological diagnosis of DD and compare skin with endomyocardial biopsy features. A control skin sample of a healthy individual was obtained from IRCCS L. Spallanzani biobank.

### 2.3. Molecular Analyses

Genomic DNA was isolated from peripheral blood using the NucleoSpin Blood (Macherey-Nagel, Duren, Germany) according to the manufacturer’s protocols. An opportunely designed custom TruSight One sequencing panel kit (Illumina, San Diego, CA, USA) was used to analyze a panel of 112 selected OMIM genes, including *ATP2A2* (MIM #108740), as well as the other 111 genes involved in cardiac disorders (Appendix A) for the list of cardiac genes included in the next-generation-sequencing [NGS] panel used for this analysis). The enriched libraries were sequenced by a NextSeq 500 instrument (Illumina, San Diego, CA, USA) (Appendix A). NGS data analysis was performed using an in-house implemented pipeline as previously described [5,6].

Bidirectional Sanger sequencing was used to validate the identified variants and perform segregation studies (Appendix A). Sanger sequencing was performed using the ABI BigDye Terminator Sequencing Kit v.3.1 (Thermo Fisher, Waltham, MA, USA) and an ABI 3130 (Thermo Fisher, Waltham, MA, USA). Primer sequences and PCR conditions are available upon request.

### 2.4. RNA Studies

Total RNA was extracted from PAXgene Blood RNA Tubes (Qiagen, Hilden, Germany) using the PAXgene Blood miRNA Kit (Qiagen, Hilden, Germany) according to the manufacturer’s instructions. cDNA was synthesized using the Superscript III first-strand synthesis kit (ThermoFisher, Waltham, MA, USA). A SYBR green-based real-time quantitative PCR (RT-qPCR) assay was used to measure *ATP2A2* mRNA (NM_001681.3) levels in peripheral blood using two primer sets spanning the junctions between exons 1–2 and 17–18, respectively (the primer sequences are available upon request). Each primer set was run in triplicate in two independent experiments using a 7900 HT Fast Real-Time PCR System (Thermo Fisher, Waltham, MA, USA). The expression ratio of the target genes was calculated by using the 2^−ΔΔCt^ method [7] and considering actin as the reference gene.

### 2.5. Protein Isolation and Western Blot

Heart tissue samples were treated as described [8]. The expression of sarcoplasmic or endoplasmic reticulum calcium (SERCA-2) antibody was visualized by using a mouse monoclonal antibody (1:100, Santa Cruz Biotechnology, Inc. Dallas, TX, USA) and Anti-α-sarcomeric actin (1:500, Sigma-Aldrich, Darmstadt, Germany) antibody was used for normalization. The signal was visualized using a secondary horseradish peroxidase-labeled goat anti-mouse antibody (goat anti-mouse IgG-HRP 1:5000, Santa Cruz Biotechnology) and enhanced chemiluminescence (ECL Clarity Biorad, Hercules, CA, USA). The purity and equal loading of the fraction were determined by measuring β-actin protein levels. Digital images of the resulting bands were quantified by the Image Lab software package, version 2.0 (Bio-Rad Laboratories, Munchen, Germany) and expressed as arbitrary densitometric units.

### 2.6. Electron Microscopy (TEM) Methods

Samples from skin biopsy, endomyocardial biopsy, and papillary cardiac muscle were rapidly fixed in glutaraldehyde 2.0% in cold buffer phosphate. After several washes in the same buffer, they were postfixed in 1.33% osmium tetroxide, then washed in buffer, dehydrated in ethanol, passed in toluene, and embedded in Epson resin following the standard schedules. Ultrathin sections stained with a uranyl substitute and lead hydroxide were observed with a JEOL 1400 plus TEM.

## 3. Results

### 3.1. Cardiac Studies

Stress ECG was stopped at 2.5 min of 25 Watts (heart rate 60/bts/min, double product 6000) because of muscle exhaustion and worsening of precordial pain associated with increased (−2 mm) deflection of ST segment in V4–V6 leads (Figure 3).

Holter monitoring (24 h) registered frequent (3200) supraventricular and polymorphic ventricular (2830) ectopic beats with some couples and triplets.

Cardiac magnetic resonance observed normal thickness of left ventricular wall (9 mm for ventricular septum and posterior wall), normal myocardial mass (64.5 g/m^2^ BSA; nv 34–70), and apical hypokinesis with depressed LV function (EF 40%) (Figure 3), while failed to show signal abnormalities in T1 and T2 mapping and LGE after gadolinium infusion.

Skin biopsy showed, both at OM and TEM, loss of adhesion with dystrophic changes of dermal cells (Figure 4), confirming the diagnosis of DD.

Likewise, endomyocardial biopsy documented at histology focal disconnection of cardiomyocytes (Figure 4, panel B) losing their spatial organization because of unendotheliolized spaces and appearing often in total disarray (Figure 4, panel B). The extent of unendotheliolized spaces was 28% higher than controls (which was 0%): see Figure 4D and Appendix A.

Affected myocytes also showed some swelling and cytoplasm vacuolization.

At the ultrastructural examination, myocyte detachment appeared to occur at cell lateral junctions (Figure 5), while intercalated disks appeared structurally well preserved. Cytoplasm vacuoles were due to the remarkable dilatation of the sarcoplasmic reticulum (Figure 5), which was the site of mutated *ATP2A2* protein and perinuclear areas of myofibrillolysis. Myocyte apoptosis was significantly higher in our patient (1639 ± 180 apoptotic nuclei) compared with normal controls 12 ± 2 (*p* < 0.05) (Figure 6).

### 3.2. Genetic Investigations

Exome sequencing on a sample from the proband identified a heterozygous G to A transition at the last nucleotide of exon 1 of *ATP2A2* gene (NM_170665.3:c.118G>A), which encodes the sarco/endoplasmic reticulum Ca(2+) ATPase type 2 isoform (SERCA2). No pathogenic variant was found in the panel of cardiac genes. Sanger sequencing confirmed segregation of the *ATP2A2* gene c.118G>A variant with DD phenotype in an affected sibling but not in two unaffected family members (Figure 3, panel A). This variant had not previously been reported in public databases (ExAC, gnomAD, 1000 Genomes Project, and NSEuroNet) and was predicted to affect splicing by in silico tools (Alamut version 2.10.0). Concordantly to the prediction, analysis of the *ATP2A2* cDNA by real-time quantitative PCR showed that c.118G>A caused 36% to 40% reduced expression of *ATP2A2* mRNA in peripheral blood leukocytes (Figure 1), consistent with *ATP2A2* c.118G>A variant being a splicing defect associated with a decreased expression of *ATP2A2* gene. Based on this evidence, the variant was classified as pathogenic.

### 3.3. Protein Studies (Western Blotting)

Protein expression of this sarcoplasmic protein failed to show significant quantitative differences compared with surgical control biopsies (Appendix A).

### 3.4. Treatment

Phosphodiesterase inhibitor aminophylline at the dosage of 200 mg bid was given to reduce the degradation of cyclic AMP and improve phosphorylation and activity of ATP2A2.

After 2 weeks, the patient manifested a disappearance of chest pain, improvement in muscle function, and reduction of cardiac arrhythmias (SV ectopic beats 280 and 180 V ectopic beats/24 h without evidence of repetitive phenomena). At control stress ECG, the patient was able to remarkably increase (2.3 times, heart rate 117/bts min., double product 14,000) physical activity without the occurrence of precordial pain while ST segment remained at the isoelectric line (Figure 2, panel H). Cardiac contractility (Left Ventricular Ejection Fraction) at 2D-echo raised to 45%.

## 4. Discussion

Although Darier disease is known as a rare disorder limited to the skin, it is likely that it may affect additional systems. Indeed, the incidence of psychiatric manifestations is so high (5) that it has been suggested that the central nervous system might be involved as well. The present study identifies for the first time a novel *ATP2A2* (c.118G>A) gene mutation affecting both skin and myocardium. These tissues exhibit similar histological and ultrastructural changes consisting of alterations of keratinocyte and cardiomyocyte cell-to-cell interactions and suggest a pathway common to both. In this regard, it is known that the function of proteins (such as connexins and tight junction proteins) responsible for lateral cell bonds are strictly Ca^++^-dependent and that disruption of Ca^++^ homeostasis can compromise cell adhesion. Pathogenic variants in the *ATP2A2* gene have been shown to cause severe disruption of Ca^++^ homeostasis by altering protein expression and/or the transport function in vitro [9,10] and lead to moderate depletion of calcium stores in keratinocytes of patients [11]. In DD, symptoms occur independently of *ATP2A2* mutation type or which aspect of SERCA2 functionality is compromised, suggesting that *ATP2A2* mutations exert their deleterious effect through haploinsufficiency, although also a dominant negative effect of the *ATP2A2* mutants has been proposed to underlie the DD phenotype. Recent studies confirm that the integrity of SERCA2 is necessary for maintaining cell adhesion [12]; its deficiency is associated with the enhancement of the MAPK pathway that can be downregulated by MEK inhibitors. In our patient, despite cardiac involvement, we cannot infer which specific aspect of SERCA2 functionality has been compromised. We proved that the *ATP2A2* c.118G>A mutation is associated with a reduction in mRNA expression through real-time quantitative PCR in peripheral blood, while the SERCA2 protein was normally expressed at Western blot assessment, suggesting a sort of posttranscriptional adaptation that upregulates the expression of the residual SERCA2 protein. At the cardiac level, mutation consequences are derangement of cardiomyocyte organization (disarray) and remarkable intracellular changes. Specifically, at the ultrastructural examination, the sarcoplasmic reticulum appeared dilated, and perinuclear areas of myofibrillolysis were commonly seen. Noteworthy, the structure of intercalated disks was preserved, confirming cell detachment occurring at the lateral cell junction. It may be argued that in a mouse model of Darier disease, there was no histological report of cardiomyocyte disconnection as well as of physiological derangements. However, it must be relieved that there was no previous endomyocardial biopsy study in humans with Darier disease and that our patient developed cardiac manifestations several decades after the appearance of skin lesions. This may suggest that cardiac involvement may occur later compared with skin damage.

Clinical implications from morphological analysis are that cardiac manifestations may occur in DD. These mimic ischemic heart disease with precordial pain worsened by effort, abnormal ECG changes and polymorphic, repetitive ventricular ectopic beats, and a decline in cardiac contractility (LVEF 40%).

Interestingly, phosphodiesterase inhibitors (PDEs), like aminophylline, which reduce the degradation of cyclic AMP and increase energy availability, may improve cardiac symptoms, abolishing chest pain and improving vital capacity through the enhancement of SERCA2 protein phosphorylation. Specifically, aminophylline competitively inhibits type 3 and type 4 PDE responsible for breaking down cyclic AMP. This may result in a stronger bond between cardiomyocytes, improving cardiac symptoms. In our patient, in addition to the disappearance of precordial pain, double product increased by 2.5 times without abnormal deflection of the ST segment of therapy stress ECG, while LVEF mildly increased to 45%.

In conclusion, specific mutations of *ATP2A2* may cause cardiac involvement in patients with DD. Cardiac pathologic changes are similar to those documented in the skin, including cardiomyocyte disconnection that promotes precordial pain, cardiac arrhythmias, and a decline in cardiac contractility.

Phosphodiesterase inhibitors that enhance SERCA2 protein phosphorylation may substantially attenuate the symptoms.

## Figures and Tables

**Figure 1 biomedicines-12-01060-f001:**
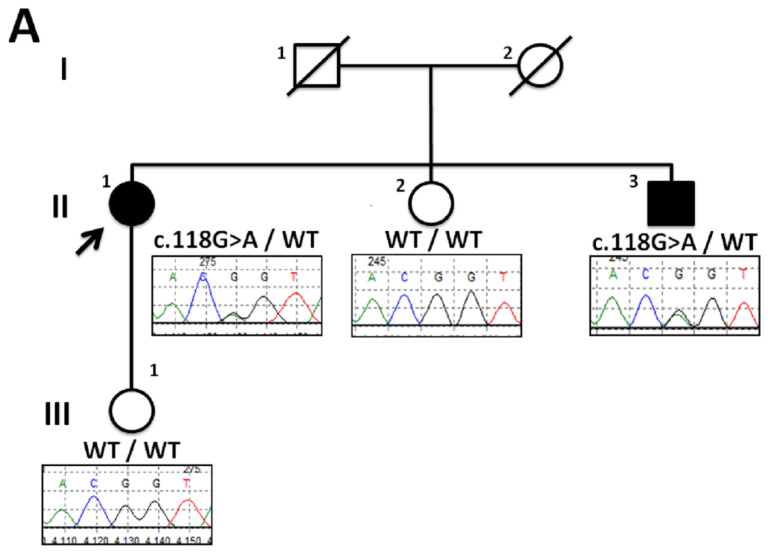
Panel (**A**)—Pedigree of DD family (proband highlighted with arrow) along with partial sequence chromatograms showing segregation of ATP2A2 mutation. The affected individuals are indicated by the solid symbol, and unaffected individuals by open symbols. Panel (**B**)—RT-qPCR ATP2A2 mRNA level quantification. Histograms show the percentages of ATP2A2 transcripts using two sets of primers designed to span the exon–exon junction between ATP2A2 exons 1–2 and 17–18, calibrated on the actin gene (ΔΔCt method) and calculated in the patient with c.118G>A mutation with respect to two wild-type healthy controls (CTRL-1 and CTRL-2). The ATP2A2 mRNA product was detected in peripheral blood at percentages ranging from 60% to 65%.

**Figure 2 biomedicines-12-01060-f002:**
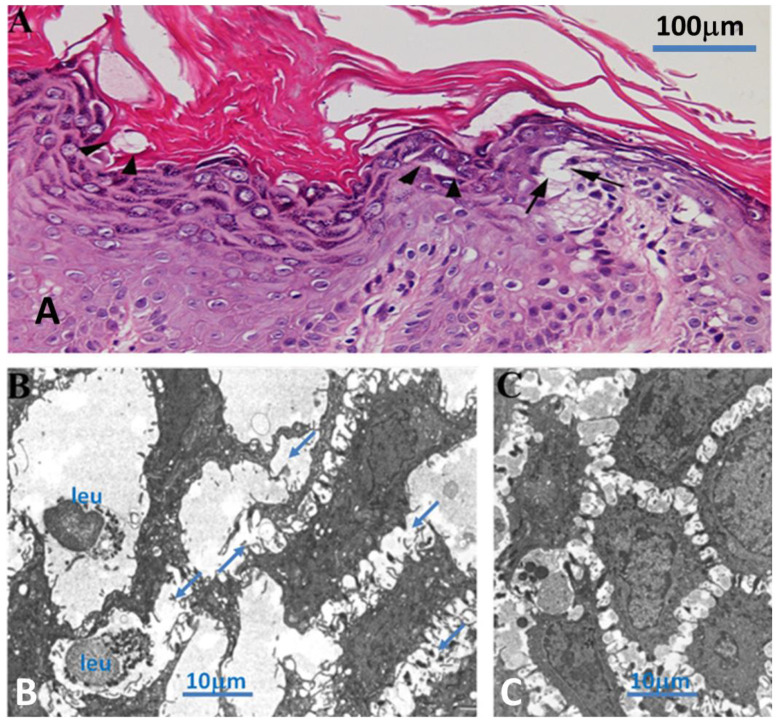
Panel (**A**): Keratinocytes disconnection in Darier disease. Characteristic round corps (arrows) are present in the upper Malpighian layer and stratum corneum. Arrowheads indicate the presence of dystrophic changes. Panel (**B**): Stratified squamous epithelium at the level of dermatologic lesion. Cell processes from keratinocytes interacting through desmosomes appear mostly disorganized. Moreover, extracellular spaces appear broadly enlarged as compared to the control skin (panel (**C**)), containing some leukocytes (left side of the figure). Arrows: disorganized junction between keratinocytes, leu = leukocyte.

**Figure 3 biomedicines-12-01060-f003:**
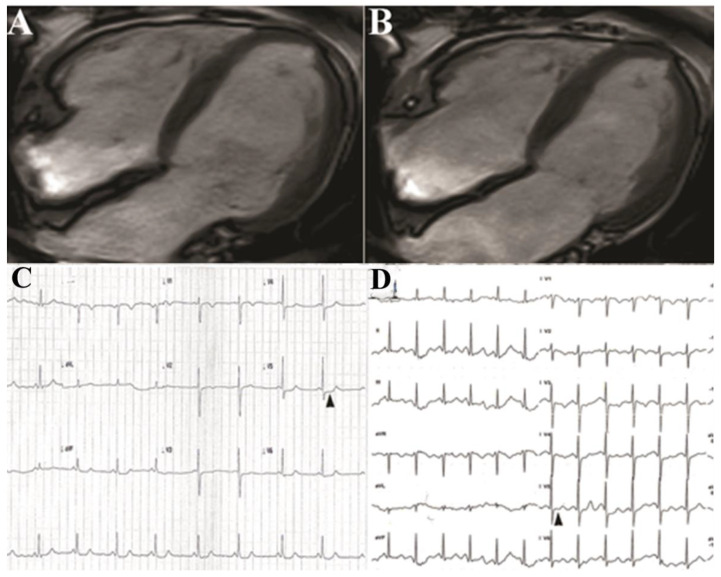
Panel (**A**,**B**): Cardiac magnetic resonance in diastole (**A**) and systole (**B**) showing reduction of left ventricular function (EF 40%). Panel (**C**): Stress ECG showing ST segment deflection in V4–V6 precordial leads associated to chest pain. Panel (**D**): Painless increase in heart rate (117 bts/min) with isoelectric ST segment, following aminophylline administration.

**Figure 4 biomedicines-12-01060-f004:**
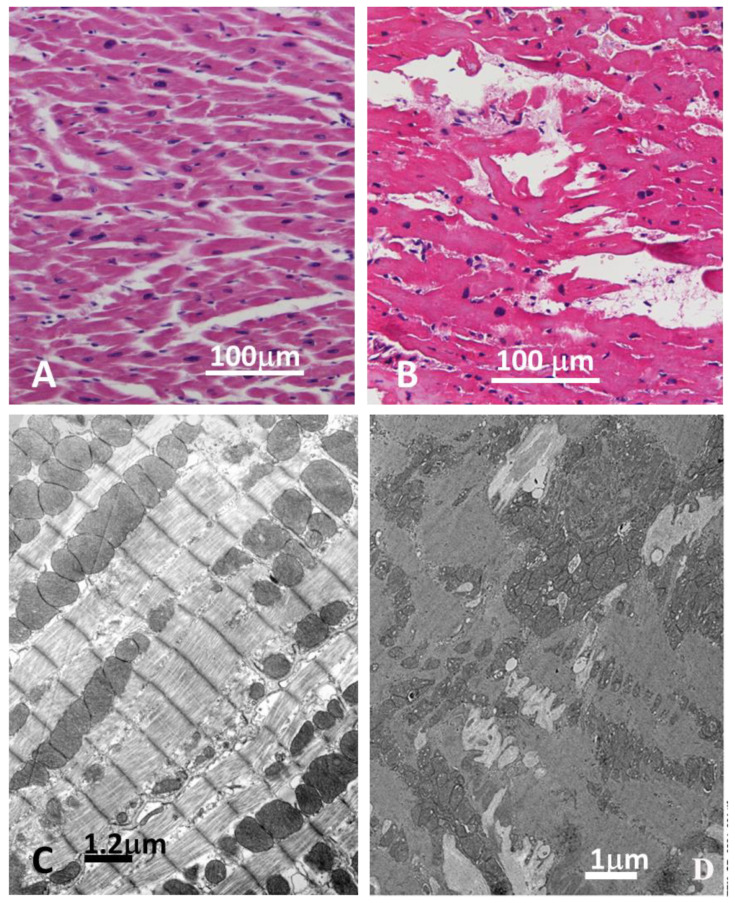
Panel (**A**): E/E stain OM micrograph of a control papillary myocardial muscle. Panel (**B**): OM similar field from patient ventricular EMB that shows normal-sized cardiomyocytes separated by unendotheliolized spaces that cause cell disarray compared with normal control (**A**). Panel (**C**): Electron microscopy of control normal papillary muscle of sample A. Panel (**D**): Transmission electro micrograph of the endomyocardial biopsy (EMB). Invaginations from extracellular spaces between cardiac myocytes from the Darier disease patient compared with normal control (**C**).

**Figure 5 biomedicines-12-01060-f005:**
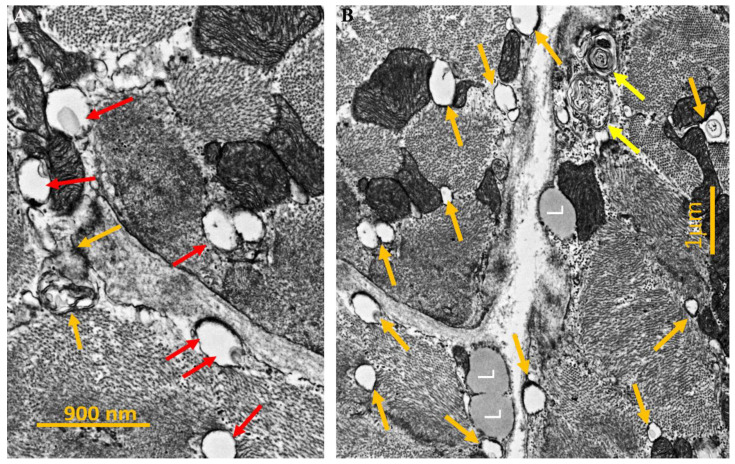
Panel (**A**)—High power magnification TEM micrograph showing dilated sarcoplasmic reticulum (SR) cisternae (red arrows). Some of them are closely associated with the mitochondrial membrane or at the level of triads (orange arrows). Spaces between myocardiocytes appear to be enlarged and with absent lateral contact. Panel (**B**)—Further details of dilated SR cisternae (orange arrows), autophagocytic vacuoles (yellow arrows), and lipid droplets (L). Spaces between myocardiocytes appear to be enlarged and with absent lateral contact.

**Figure 6 biomedicines-12-01060-f006:**
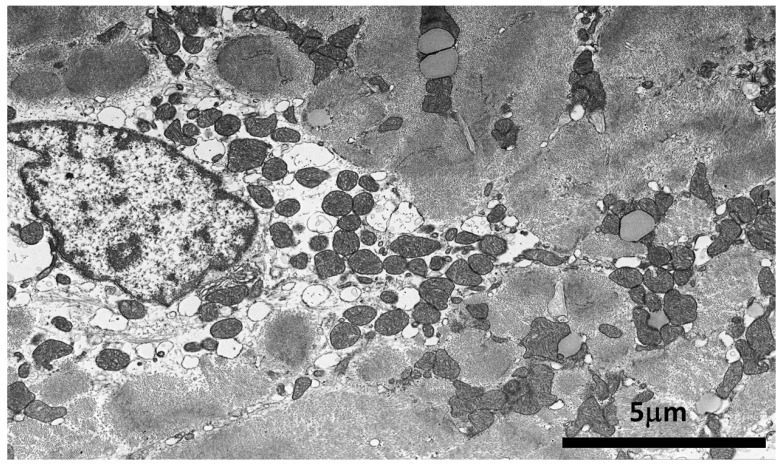
Detail of a cardiomyocyte from patient EMB with the most frequent ultrastructural nuclear and perinuclear alterations. Nuclear shape and chromatin aggregation are typical in early changes of apoptosis. Perinuclear region shows myofibrillolysis, dilated SR cisternae, and still-preserved mitochondria.

## Data Availability

The datasets used and analyzed during the current study are available from the corresponding authors upon reasonable request.

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
