# Peer review of "Novel ATP2A2 Gene Mutation c.118G>A Causing Keratinocyte and Cardiomyocyte Disconnection in Darier Disease"

_biomedicines, 2024, doi:10.3390/biomedicines12051060_

Round 1
Reviewer 1 Report
Comments and Suggestions for Authors
The authors described the case of a patient with rare Darier disease who was diagnosed with a new mutation of the ATP2A2 gene. The description is comprehensive and includes, among others: endomyocardial biopsy, skin biopsy and DNA,RNA and Western Blot analysis. Moreover, the authors describe the effects of aminophyline treatment. The presented paper is of clinical significance and may be published in the journal but it should be mention that is one case so we should suppose the meanings of described changes
Author Response
Reviewer 1
(x) I would not like to sign my review report
( ) I would like to sign my review report
Quality of English Language
(x) I am not qualified to assess the quality of English in this paper
( ) English very difficult to understand/incomprehensible
Yes Can be improved Must be improved Not applicable
Does the introduction provide sufficient background and include all relevant references?
(x) ( ) ( ) ( )
Are all the cited references relevant to the research?
(x) ( ) ( ) ( )
Is the research design appropriate?
(x) ( ) ( ) ( )
Are the methods adequately described?
(x) ( ) ( ) ( )
Are the results clearly presented?
(x) ( ) ( ) ( )
Are the conclusions supported by the results?
( ) ( ) (x) ( )
Comments and Suggestions for Authors
The authors described the case of a patient with rare Darier disease who was diagnosed with a new mutation of the ATP2A2 gene. The description is comprehensive and includes, among others: endomyocardial biopsy, skin biopsy and DNA,RNA and Western Blot analysis. Moreover, the authors describe the effects of aminophyline treatment. The presented paper is of clinical significance and may be published in the journal but it should be mention that is one case so we should suppose the meanings of described changes
Reply: We sincerely thank the reviewer for His/Her considerations.
Reviewer 2 Report
Comments and Suggestions for Authors
In the reviewed manuscript, the authors Frustaci et al. report case study – patient suffering from Darier Disease (DD), where beyond classical symptoms cardiac complications have occured. In general, this study represents very novel and unique work, as the cardiac complications in the DD may indeed appear in very late stage of the disease. On the other site, it is indeed surprising while keeping in mind the SERCA2 protein is crucial for the cardiac myocyte calcium handling, the patients demonstrate solely skin manifestations and cardiac complications are practically absent. Therefore this case report has high potential to open the new basic research as well as diagnostic and therapeutic avenues for DD in the future. Moreover, presented case report is of high importance when keeping in mind these results, i.e. very late stage of the disease is pretty hard to document in an animal or cellular model. In addition, the authors obtained the left ventricular myocardium samples from the patient and obtained similar clinical material from healthy controls, which is indeed a challenging task. Therefore this manuscript in general represents new insight to the field of research of DD with unique original results. Authors have used robust, well established methods to obtain presented results, performing basic characterization of the patient including family history, mRNA, Western Blotting, in vivo cardiac studies as well as histology and very importantly ultrastructural examination via transmission electron microscopy (TEM).
Specifically in this case, the authors have found new mutation of the ATP2A2 gene causing beyond standard skin lesions palpitations and chest pain at rest pointing out for myocardial dysfunction in reported elderly patient. The authors propose significant changes in the patient left ventricular myocytes, namely cardiac myocyte separation specifically in lateral junctions as well as dilatation of sarcoplasmic reticulum. Even more importantly, by medical treatment of the patient with a phosphodiesterase inhibitor enhancing SERCA2 protein phosphorylation, the aminophylline, the clinical status of the patient was remarkably improved.
Although I personally consider this case report as highly important not only for the field of the DD, but for general understanding of cardiac muscle function and putative improvement in pharmacological therapy as well, what is clear from all the reasoning above, there are several crucial points which have to be addressed to definitely prove the conclusions provided by the authors, see below.
Major issues (sorted by relevance):
1. Disconnection of cardiac myocytes in the patient myocardium is depicted already in the title of the manuscript. However, this conclusion is not well supported by provided data. First, in the histological images (Fig. 2 C,D) control group is missing, moreover a quantitative analysis of the unendotheliolized spaces between the myocytes would improve this evidence a lot. Second, similar approach should be applied for the TEM images – in the Fig. 2 E,F there is only one intercellular space shown. Moreover, TEM image from control is in different scale as the patient sample. Here I strongly recommend to implement several representative TEM images both from the patient and control demonstrating the altered cardiomyocyte separation at lateral junctions, but not at the intercalated disks.
2. Line 134-136: this sentence should be documented by solid evidence, TEM images showing vacuolization of the SR should be implemented from the patient and control.
3. Control group, especially for the TEM image and myocyte apoptosis data should be described in detail in the Methods section.
4. Lines 147-148 – it is not clear what the expression ‘this sarcoplasmic protein’ means. The WB including the housekeeper protein should be presented in the Results section.
5. Supplementary data are missing the manuscript.
6. The Introduction and especially Discussion section would benefit from short description of beneficial effects of the phosphodiesterase inhibitors to cardiac muscle in clinics and in experimental condition.
Minor issues:
I strongly recommend to organize the Figures in following way: as first Fig. 3, second Fig. 1 and third Fig. 2.
Lines 184-185 – what exact parameter(s) the term ‘Cardiac contractility’ means in this statement.
Fig. 2A – the arrows are described in the Fig.2 legend, however, the arrowheads are not. Please clarify.
Lines 218-220 – it is not clear, if this paragraph describes the findings from this manuscript. In that case use past simple tense. Otherwise add a reference(s) for these statements.
Fig. 2. In C,D – scale bars are missing. In E,F – the fonts for the scale bar markings are too small.
Fig. 1A – scale bar is missing
The manuscript may benefit from brief declaring the fact DD belongs to rare diseases group
Typographical errors:
Line 62 – correct ‘Electronmicroscopy’ to ‘Electron microscopy’
Line 63 – correct ‘myocardiocytes’ to ‘cardiac myocytes’
Line 69 – I recommend to correct ‘They included’ to ‘We performed’
Line 140 – I recommend to omit ‘(A)’ in this line
Line 196 – correct ‘CA++’ to ‘Ca++’
Comments on the Quality of English LanguageSee comments in the body of the review.
Author Response
Reviewer 2
(x) I would not like to sign my review report
( ) I would like to sign my review report
Quality of English Language
(x) Minor editing of English language required
( ) English language fine. No issues detected
Yes Can be improved Must be improved Not applicable
Does the introduction provide sufficient background and include all relevant references?
( ) (x) ( ) ( )
Are all the cited references relevant to the research?
(x) ( ) ( ) ( )
Is the research design appropriate?
(x) ( ) ( ) ( )
Are the methods adequately described?
( ) (x) ( ) ( )
Are the results clearly presented?
( ) (x) ( ) ( )
Are the conclusions supported by the results?
( ) (x) ( ) ( )
Comments and Suggestions for Authors
In the reviewed manuscript, the authors Frustaci et al. report case study – patient suffering from Darier Disease (DD), where beyond classical symptoms cardiac complications have occured. In general, this study represents very novel and unique work, as the cardiac complications in the DD may indeed appear in very late stage of the disease. On the other site, it is indeed surprising while keeping in mind the SERCA2 protein is crucial for the cardiac myocyte calcium handling, the patients demonstrate solely skin manifestations and cardiac complications are practically absent. Therefore this case report has high potential to open the new basic research as well as diagnostic and therapeutic avenues for DD in the future. Moreover, presented case report is of high importance when keeping in mind these results, i.e. very late stage of the disease is pretty hard to document in an animal or cellular model. In addition, the authors obtained the left ventricular myocardium samples from the patient and obtained similar clinical material from healthy controls, which is indeed a challenging task. Therefore this manuscript in general represents new insight to the field of research of DD with unique original results. Authors have used robust, well established methods to obtain presented results, performing basic characterization of the patient including family history, mRNA, Western Blotting, in vivo cardiac studies as well as histology and very importantly ultrastructural examination via transmission electron microscopy (TEM).
Reply: We sincerely thank the reviewer for His/Her considerations.
Specifically in this case, the authors have found new mutation of the ATP2A2 gene causing beyond standard skin lesions palpitations and chest pain at rest pointing out for myocardial dysfunction in reported elderly patient. The authors propose significant changes in the patient left ventricular myocytes, namely cardiac myocyte separation specifically in lateral junctions as well as dilatation of sarcoplasmic reticulum. Even more importantly, by medical treatment of the patient with a phosphodiesterase inhibitor enhancing SERCA2 protein phosphorylation, the aminophylline, the clinical status of the patient was remarkably improved.
Although I personally consider this case report as highly important not only for the field of the DD, but for general understanding of cardiac muscle function and putative improvement in pharmacological therapy as well, what is clear from all the reasoning above, there are several crucial points which have to be addressed to definitely prove the conclusions provided by the authors, see below.
Major issues (sorted by relevance):
- Disconnection of cardiac myocytes in the patient myocardium is depicted already in the title of the manuscript. However, this conclusion is not well supported by provided data. First, in the histological images (Fig. 2 C,D) control group is missing, moreover a quantitative analysis of the unendotheliolized spaces between the myocytes would improve this evidence a lot. Second, similar approach should be applied for the TEM images – in the Fig. 2 E,F there is only one intercellular space shown. Moreover, TEM image from control is in different scale as the patient sample. Here I strongly recommend to implement several representative TEM images both from the patient and control demonstrating the altered cardiomyocyte separation at lateral junctions, but not at the intercalated disks.
- Reply : We thank the reviewer for the points raised that further help to clarify the content of the study. (Figures’ position has been modified accordingly with reviewer 4)
Specifically :
- a control histologic picture of normal myocardium without unendetheliolized spaces is provided as comparison (Fig 3, panel C, Fig S4).
- Unendotheliolazed areas in the pt myocardium have been morphometrically quantified and they were found to be higher by 28% compared with normal control ( 0%).
- New TEM images are provided even at low magnification, as supplemental material to demonstrate a cardiomyocytes disconnection at lateral junction. Fig S5
- Line 134-136: this sentence should be documented by solid evidence, TEM images showing vacuolization of the SR should be implemented from the patient and control.
Reply: A new ultrastructural image (Fig S6) showing dilatation of sarcoplasmic reticulum where the abnormal protein is located is now available in the supplemental material.
- Control group, especially for the TEM image and myocyte apoptosis data should be described in detail in the Methods section.
Reply : Description of methods for assessment of myocyte apoptosis is now reported in the methods section.
- Lines 147-148 – it is not clear what the expression ‘this sarcoplasmic protein’ means. The WB including the housekeeper protein should be presented in the Results section.
Reply: A new figure with WB of SERCA2A has been added as supplemental material including the housekeeper protein.
- Supplementary data are missing the manuscript.
Reply: Supplementary data are now provided.
- The Introduction and especially Discussion section would benefit from short description of beneficial effects of the phosphodiesterase inhibitors to cardiac muscle in clinics and in experimental condition.
Reply: the mechanism of beneficial effect of phosphodiesterase inhibitors presumed in our pt is mentioned in discussion.
Minor issues:
I strongly recommend to organize the Figures in following way: as first Fig. 3, second Fig. 1 and third Fig. 2.
Reply: The order of figures has been changed accordingly.
Lines 184-185 – what exact parameter(s) the term ‘Cardiac contractility’ means in this statement.
Reply : For cardiac contractility we intend Left Ventricular Ejection Fraction which raised mildly to 45%
Fig. 2A – the arrows are described in the Fig.2 legend, however, the arrowheads are not. Please clarify.
Reply: Arrowheads indicate presence of dystrophic changes
Lines 218-220 – it is not clear, if this paragraph describes the findings from this manuscript. In that case use past simple tense. Otherwise add a reference(s) for these statements.
Reply: Done.
Fig. 2. In C,D – scale bars are missing. In E,F – the fonts for the scale bar markings are too small.
Reply: done
The manuscript may benefit from brief declaring the fact DD belongs to rare diseases group
Typographical errors:
Line 62 – correct ‘Electronmicroscopy’ to ‘Electron microscopy’
Reply: Done.
Line 63 – correct ‘myocardiocytes’ to ‘cardiac myocytes’
Reply: Done.
Line 69 – I recommend to correct ‘They included’ to ‘We performed’
Reply: Done.
Line 140 – I recommend to omit ‘(A)’ in this line
Reply: Done.
Line 196 – correct ‘CA++’ to ‘Ca++
Reply: Done.
Reviewer 3 Report
Comments and Suggestions for Authors
The manuscript “Novel ATP2A2 Gene Mutation c.118G>A Causing Keratinocyte and Cardiomyocyte Disconnection in Darier Disease” by Andrea Frustaci et al is a kind of case report claiming that novel mutation in ATP2A2 gene underlies the heart problems of a 62-years old lady. The manuscript is small in volume, but raises many questions..
1. The description of the patient’s condition and his anamnesis are presented unsatisfactorily. The reader cannot glean from it information about the patient’s body weight, body mass index, the presence/absence of concomitant diseases (diabetes, alcoholism, drug addiction, other chronic pathologies and addictions), the time of onset and the dynamics of development of heart failure symptoms..
2. The authors say that the morphological, dynamic, ultrastructural features of the patient's heart and biopsy samples differed from “normal control”, but do not explain what this term means. Were these 62-year-old women with the same set of comorbidities and habits, except Darier's disease?
3. The authors report that “endomyocardial biopsy documented at histology disconnection of cardiomyocytes (Fig 2, panel C), loosing their spatial organization because of unendotheliolized spaces and appearing often in total disarray (Fig 2, panel D). Affected myocytes also showed some swelling and cytoplasm vacuolization.” However, it is completely unclear how reliable these changes are, how they are expressed quantitatively, what percentage of tissues have these abnormalities, and how the histological characteristics were affected by treatment by aminophylline.
4. The authors showed that the level of mutated mRNA in peripheral blood was reduced by 40%, while WB analysis did not reveal differences in the level of SERCA2 (these data were not presented. Why?). Therefore, it is difficult to understand how the effect of reducing the ATP2A2 gene transcript level is realized. In addition, the authors do not provide any evidence that the activity, stability, or sensitivity to activators of SERCA2 is in any way altered in the patient.
5. The authors report that a two-week course of aminophylline caused normalization of cardiac activity. However, this may be due to many cAMP- and cGMP-dependent proteins, including, for example, RyR2, a channel involved in cardiac muscle contraction.
6. Minor point. Figures are on the wrong places.
To conclude, the authors' statement that the mutation they discovered causes cardiomyocyte disconnection, which may be associated with cardiac dysfunction, is currently not supported by a sufficient set of experimental data.
Author Response
REVIEWER 3
(x) I would not like to sign my review report
( ) I would like to sign my review report
Quality of English Language
(x) I am not qualified to assess the quality of English in this paper
Does the introduction provide sufficient background and include all relevant references?
( ) ( ) (x) ( )
Are all the cited references relevant to the research?
( ) (x) ( ) ( )
Is the research design appropriate?
( ) ( ) (x) ( )
Are the methods adequately described?
( ) ( ) (x) ( )
Are the results clearly presented?
( ) ( ) (x) ( )
Are the conclusions supported by the results?
( ) ( ) (x) ( )
Comments and Suggestions for Authors
The manuscript “Novel ATP2A2 Gene Mutation c.118G>A Causing Keratinocyte and Cardiomyocyte Disconnection in Darier Disease” by Andrea Frustaci et al is a kind of case report claiming that novel mutation in ATP2A2 gene underlies the heart problems of a 62-years old lady. The manuscript is small in volume, but raises many questions..
- The description of the patient’s condition and his anamnesis are presented unsatisfactorily. The reader cannot glean from it information about the patient’s body weight, body mass index, the presence/absence of concomitant diseases (diabetes, alcoholism, drug addiction, other chronic pathologies and addictions), the time of onset and the dynamics of development of heart failure symptoms..
- Reply: Anamnestic details are now provided in the case study, as requested.
- The authors say that the morphological, dynamic, ultrastructural features of the patient's heart and biopsy samples differed from “normal control”, but do not explain what this term means. Were these 62-year-old women with the same set of comorbidities and habits, except Darier's disease?
Reply: Histological and ultrastructural differences from our pt and normal control are now remarked consisting:
- Presence of cardiomyocytes hypertrophy with some degenerative changes of cytoplasm and dilatation of sarcoplasmic reticulum (Site of mutated protein) at electron microscopy.
- Cell disconnection at lateral junction of cardiomyocytes see Figure S5) that at morphometry is associated with an increase of unendotheliolized areas by 28% compared with normal tissue (0%)
- This structural abnormalities are associated with cardiac dilatation, reduction of left ventricular contractility (LVEF 40%), and occurrence of ventricular arrhythmias.
- The authors report that “endomyocardial biopsy documented at histology disconnection of cardiomyocytes (Fig 2, panel C), loosing their spatial organization because of unendotheliolized spaces and appearing often in total disarray (Fig 2, panel D). Affected myocytes also showed some swelling and cytoplasm vacuolization.” However, it is completely unclear how reliable these changes are, how they are expressed quantitatively, what percentage of tissues have these abnormalities, and how the histological characteristics were affected by treatment by aminophylline.
Reply: We do believe that phosphodiesterase inhibitors may have acted increasing ATP production. We of course we are not provided of a control biopsy to document a structural modification.
- The authors showed that the level of mutated mRNA in peripheral blood was reduced by 40%, while WB analysis did not reveal differences in the level of SERCA2 (these data were not presented. Why?). Therefore, it is difficult to understand how the effect of reducing the ATP2A2 gene transcript level is realized. In addition, the authors do not provide any evidence that the activity, stability, or sensitivity to activators of SERCA2 is in any way altered in the patient.
Reply: Data on WB are now shown as supplemental material.
- The authors report that a two-week course of aminophylline caused normalization of cardiac activity. However, this may be due to many cAMP- and cGMP-dependent proteins, including, for example, RyR2, a channel involved in cardiac muscle contraction.
Reply: Of course this is just an hypothesis that is not demonstrated yet.
- Minor point. Figures are on the wrong places.
Reply: Figures position has been more appropriately modified.
To conclude, the authors' statement that the mutation they discovered causes cardiomyocyte disconnection, which may be associated with cardiac dysfunction, is currently not supported by a sufficient set of experimental data.
Reply: We do believe the newer histological, ultrastructural and morphometric data provide robust evidence on the relation between ATP2A2 gene mutation and cardiac damage.
Reviewer 4 Report
Comments and Suggestions for Authors
This work is very interesting: the authors describe a clinical case of a 62-year-old patient developing Darier disease as a result of a novel c.118G>A mutation in the ATP2A2 gene in the heterozygous state, which probably leads to a splicing disorder. This work expands our understanding of pathologic single nucleotide polymorphisms, and the results may improve current genetic tests in many countries. I have only minor comments.
1. References to articles should be formatted according to MDPI rules. That is, they should be enclosed in square colored brackets and bolded.
2. It should be done so that the Figure 1 comes before the Figure 2.
3. The molecular mechanism of Darier disease development and how the ATP2A2 gene is involved needs to be described in more detail.
4. Please briefly describe the mechanism of disease manifestation in case of other mutations and their prevalence in the population, if such data are available.
5. Does this case of Darier disease differ in clinical manifestations from the previously described ones? Is there evidence that Darier disease caused by different mutations in the ATP2A2 gene may have different phenotypic manifestations?
Comments on the Quality of English Language
English is OK.
Author Response
REVIEWER 4
(x) I would not like to sign my review report
( ) I would like to sign my review report
Quality of English Language
(x) Minor editing of English language required
Does the introduction provide sufficient background and include all relevant references?
( ) (x) ( ) ( )
Are all the cited references relevant to the research?
(x) ( ) ( ) ( )
Is the research design appropriate?
(x) ( ) ( ) ( )
Are the methods adequately described?
(x) ( ) ( ) ( )
Are the results clearly presented?
( ) (x) ( ) ( )
Are the conclusions supported by the results?
(x) ( ) ( ) ( )
Comments and Suggestions for Authors
This work is very interesting: the authors describe a clinical case of a 62-year-old patient developing Darier disease as a result of a novel c.118G>A mutation in the ATP2A2 gene in the heterozygous state, which probably leads to a splicing disorder. This work expands our understanding of pathologic single nucleotide polymorphisms, and the results may improve current genetic tests in many countries. I have only minor comments.
- References to articles should be formatted according to MDPI rules. That is, they should be enclosed in square colored brackets and bolded.
Reply: Done!
- It should be done so that the Figure 1 comes before the Figure 2.
Reply: Thank you for the observation : figures’ position has been modified accordingly.
- The molecular mechanism of Darier disease development and how the ATP2A2 gene is involved needs to be described in more detail.
Reply: Extensive description of molecular alterations following ATP2A2 gene mutation , are reported in the introduction.
- Please briefly describe the mechanism of disease manifestation in case of other mutations and their prevalence in the population, if such data are available.
Reply: Genetic and clinical characteristics are now reported in the introduction section.
- Does this case of Darier disease differ in clinical manifestations from the previously described ones? Is there evidence that Darier disease caused by different mutations in the ATP2A2 gene may have different phenotypic manifestations?
Reply: This is the first description of ATP2A2 gene mutation involving cardiac structure and function.
Round 2
Reviewer 2 Report
Comments and Suggestions for Authors
The authors, based on reviewers’ comments, have made significant improvement of the manuscript. However, several details have to be implemented to reach the quality required by the Biomedicines Journal, see below:
1. The calibration bars for the histology images in Fig. 3C and 3D (former Fig. 2C and 2D) are still missing.
2. Although the authors have provided additional figures in the Supplementary data (Fig. S4), the main comparison for Fig. 3E (former Fig. 2E) and Fig. 3F is not acceptable: Fig. 3E is in much lower magnification (1 um) compared to Fig. 3F which is in much higher magnification. Both Fig. 3E and Fig. 3F should be in the same scale. Moreover, in the current version of the manuscript, the Fig. 3F the scale bar indicates 100 nm, however in the original for the same image there was information of 300 nm provided.
3. Basic characterization (age, sex, possible medication) of the control subject/donor (data on Fig. 3D and 3F; Fig. S4) is missing in the Method section.
Author Response
Reviewer 2
Comments and Suggestions for Authors
The authors, based on reviewers’ comments, have made significant improvement of the manuscript. However, several details have to be implemented to reach the quality required by the Biomedicines Journal, see below:
- The calibration bars for the histology images in Fig. 3C and 3D (former Fig. 2C and 2D) are still missing.
Reply: Done.
- Although the authors have provided additional figures in the Supplementary data (Fig. S4), the main comparison for Fig. 3E (former Fig. 2E) and Fig. 3F is not acceptable: Fig. 3E is in much lower magnification (1 um) compared to Fig. 3F which is in much higher magnification. Both Fig. 3E and Fig. 3F should be in the same scale. Moreover, in the current version of the manuscript, the Fig. 3F the scale bar indicates 100 nm, however in the original for the same image there was information of 300 nm provided.
Reply: The suggested changed have been obtained for the figures.
- Basic characterization (age, sex, possible medication) of the control subject/donor (data on Fig. 3D and 3F; Fig. S4) is missing in the Method section.
Reply: data on control have been introduced.
Reviewer 3 Report
Comments and Suggestions for Authors
The revised version of the manuscript “Novel ATP2A2 Gene Mutation c.118G>A Causing Keratinocyte and Cardiomyocyte Disconnection in Darier Disease” by Andrea Frustaci et al seems definitely improved variant. Authors provided several explanatory paragraphs, details of the patient anamnesis, and supplementary figures. Some of these figures, namely Figs. S5 and S6, deserve to be included in the main text. I am not completely satisfied with the Author’s response; however, presumably I stated questions insufficiently clear. At this stage I can accept Author’s interpretation if they do not insist that “A Novel ATP2A2 Gene Mutationgene variant c.118G>A, causing keratinocyte and cardiomyocyte disconnection in Darier disease, is reported” (Abstract). I can agree that “Pathogenic variants of ATP2A2 gene MAY cause cardiac involvement in patients with DD ” (Abstract). Although, based on the data presented, the most likely cause of the observed cardiac cytopathology appears to be ER stress caused by misfolded proteins, this is to be proven.
Comment.
The authors try to minimize the volume of text, often to the detriment of coherence, clarity, and persuasiveness of the information. In particular, the number of references in the introduction and discussion is only 11, although many statements require appropriate citations. In addition, the methods used are described sparingly, and data on the antibodies used are not provided. This approach directly contradicts the rules for Authors.
Article: These are original research manuscripts. The work should report scientifically sound experiments and provide a substantial amount of new information. The article should include the most recent and relevant references in the field. The structure should include an Abstract, Keywords, Introduction, Materials and Methods, Results, Discussion, and Conclusions (optional) sections.
Author Response
Comments and Suggestions for Authors
The revised version of the manuscript “Novel ATP2A2 Gene Mutation c.118G>A Causing Keratinocyte and Cardiomyocyte Disconnection in Darier Disease” by Andrea Frustaci et al seems definitely improved variant. Authors provided several explanatory paragraphs, details of the patient anamnesis, and supplementary figures. Some of these figures, namely Figs. S5 and S6, deserve to be included in the main text. I am not completely satisfied with the Author’s response; however, presumably I stated questions insufficiently clear. At this stage I can accept Author’s interpretation if they do not insist that “A Novel ATP2A2 Gene Mutationgene variant c.118G>A, causing keratinocyte and cardiomyocyte disconnection in Darier disease, is reported” (Abstract). I can agree that “Pathogenic variants of ATP2A2 gene MAY cause cardiac involvement in patients with DD ” (Abstract). Although, based on the data presented, the most likely cause of the observed cardiac cytopathology appears to be ER stress caused by misfolded proteins, this is to be proven.
Reply: The sentence ‘’ “Pathogenic variants of ATP2A2 gene MAY cause cardiac involvement in patients with DD ” has now been included in the abstract! Figures 5 and 6 have been included in the main text.
Comment.
The authors try to minimize the volume of text, often to the detriment of coherence, clarity, and persuasiveness of the information. In particular, the number of references in the introduction and discussion is only 11, although many statements require appropriate citations. In addition, the methods used are described sparingly, and data on the antibodies used are not provided. This approach directly contradicts the rules for Authors.
Reply: Methods are now described in details.
Article: These are original research manuscripts. The work should report scientifically sound experiments and provide a substantial amount of new information. The article should include the most recent and relevant references in the field. The structure should include an Abstract, Keywords, Introduction, Materials and Methods, Results, Discussion, and Conclusions (optional) sections.
Reply: a correct sequence is now present.